# Effects of 450 MeV Kr Swift Heavy Ion Irradiation on GaN-Based Terahertz Schottky Barrier Diodes

**DOI:** 10.3390/mi16030288

**Published:** 2025-02-28

**Authors:** Yan Ren, Yongtao Yu, Shengze Zhou, Chao Pang, Yinle Li, Zhifeng Lei, Hong Zhang, Zhihong Feng, Xubo Song, Honghui Liu, Yongli Lou, Yiqiang Ni

**Affiliations:** 1School of Reliability and Systems Engineering, Beihang University, Beijing 100191, China; xie_sk@126.com; 2China Electronic Product Reliability and Environmental Testing Research Institute (CEPREI), Guangzhou 510640, China; yuyt2010@163.com (Y.Y.); zhousze@163.com (S.Z.); pangchao@ceprei.com (C.P.); liyinle@ceprei.com (Y.L.); leizhifeng@ceprei.com (Z.L.); zhanghong1@ceprei.com (H.Z.); liuhh@ceprei.com (H.L.); 3National Key Laboratory of Solid-State Microwave Devices and Circuits, Hebei Semiconductor Research Institute, Shijiazhuang 050051, China; ga917vv@163.com (Z.F.); sunny1989.song@gmail.com (X.S.)

**Keywords:** GaN-based THz SBDs, heavy ions irradiation, trap states, low-frequency noise, photoluminescence

## Abstract

GaN-based terahertz (THz) Schottky barrier diodes (SBDs) are critical components for achieving high-power performance in THz frequency multipliers. However, the space applications of GaN-based THz SBDs are significantly constrained due to insufficient research on the effects of space irradiation. This work investigates the effects of 450 MeV Kr swift heavy ion (SHI) irradiation on the electrical characteristics and induced defects in GaN-based THz SBDs. It was found that the high-frequency performance of GaN-based THz SBDs is highly sensitive to Kr SHI irradiation, which can be attributed to defects induced in the GaN epitaxial layer by the irradiation. Low-frequency noise analysis reveals trap states located at an energy level of approximately 0.62 eV below the conduction band. Moreover, the results from SRIM calculation and photoluminescence spectra confirmed the presence of irradiation-induced defects caused by Kr SHI irradiation.

## 1. Introduction

Terahertz (THz) technology has rapidly advanced due to its potential applications in astrophysics, earth observation, radio astronomy, remote sensing, and inter-satellite communication [1,2,3,4,5,6]. Currently, the output power of GaAs-based THz frequency multipliers is proving increasingly unable to satisfy the demands of THz systems, which is attributed to the relatively low breakdown voltage (*V_br_*) of GaAs material [7,8,9]. Compared with GaAs, GaN-based THz frequency multipliers have achieved higher output power owing to their superior *V_br_* and enhanced thermal conductivity [9,10,11,12,13,14,15], which is a promising technology for the development of high-power THz multipliers in space applications.

The output power of THz frequency multipliers is dependent on the nonlinear functionality provided by THz Schottky barrier diodes (SBDs) within the circuit integrated into a waveguide [7,14,16]. However, there is limited research on the radiation damage characteristics of GaN-based THz SBDs, which severely restricts the space applications of GaN-based THz frequency multipliers.

GaN-based THz SBDs must resist irradiation from protons, swift heavy ions (SHIs), gamma rays, and neutrons to meet the requirements of space environment applications [17,18]. Despite constituting only approximately 1% of the space environment, SHIs are the primary cause of the most significant damage to electronic components. Compared with protons and neutrons, SHIs excite and ionize target atoms significantly more along their trajectories within devices, such as total ionizing dose, single-event effects (SEEs), and displacement damage [19]. According to research reports on the radiation effects of GaN-based devices, these devices exhibit exceptionally strong resistance to total ionizing dose effect [20,21]. The SEEs induced by SHIs are more prominent in GaN-based power devices, whereas they are nearly unobservable in low-voltage or RF devices [22]. Therefore, for GaN-based THz SBD devices, the displacement damage effect induced by SHIs is a key factor in space irradiation [23]. SHIs passing through the device generate defects such as gaps, vacancies, and complexes [19], which can accumulate into defect clusters and even form potential ion tracks, leading to the degradation of device performance.

For SHI irradiation of GaN-based devices, researchers primarily analyze the degradation of electrical and material properties caused by radiation damage [24,25,26,27,28,29,30]. These studies typically focus on the microscopic mechanisms caused by SHIs within the device, while less attention is given to the effects of SHI irradiation on the frequency and power characteristics of the device [17,19]. However, GaN-based devices are precisely advantageous for high-frequency and high-power applications. Therefore, this paper reports the effects of 450 MeV Kr SHI irradiation on key parameters such as the cut-off frequency (*f_c_*) and reverse leakage (*I_R_*) of GaN-based THz SBDs. It was observed that Kr SHI irradiation degraded the high-frequency performance of GaN-based THz SBDs, which was attributed to defects introduced into the GaN epitaxial layer. Trap states were identified via low-frequency noise (LFN) spectra analysis, while photoluminescence (PL) spectra analysis confirmed the presence of irradiation-induced defects resulting from Kr SHI irradiation.

## 2. Experimental Procedure

The irradiation damage in GaN is determined by a complex interplay between irradiation parameters (type, energy, and dose) and material properties (carrier density, impurity content, and dislocation density), such as the carrier removal effect due to trapping of carriers into radiation-induced defects varies depending on the growth method of GaN layers [18]. Metal–organic chemical vapor deposition (MOCVD) technology is a widely used technique for epitaxial GaN growth, which is attributable to low inherent defect density, appropriate epitaxial growth rate, and cost-effectiveness. The low intrinsic defect density of GaN materials grown using this method can effectively mitigate the carrier removal effect caused by irradiation. The designed GaN-based THz SBD is shown in Figure 1. The GaN materials used in this work were grown on a 2-inch silicon carbide (SiC) substrate by MOCVD. The GaN epitaxial layers consist of a 450 nm high-resistance (HR) buffer layer, a 2.1 μm n^+^-GaN layer (doping concentration of 10^19^ cm^−3^), and a 150 nm n^−^-GaN layer (with a doping concentration of 5 × 10^17^ cm^−3^). Fabrication of GaN-based THz SBD mainly involves four steps: (1) GaN materials were dry etched onto SiC substrates to achieve mesa isolation using inductively coupled plasma (ICP); (2) the cathode metals of Ti/Al/Ni/Au (20/120/50/80 nm) were deposited on n^+^-GaN layer by electron-beam evaporation (EBE), followed by rapid thermal annealing in the N_2_ environment to form the ohmic contact; (3) the anode metals of Ni/Au (50/100 nm) were deposited on the n^−^-GaN layer by EBE to form the Schottky contact, with an anode diameter of 6 μm; (4) the gold air bridge (thickness of 1 μm) was achieved by the electroplating technique, as shown in Figure 1b,c.

The SHI irradiation experiment was conducted at the Space Environment Simulation Research Infrastructure (SESRI), Harbin Institute of Technology. The GaN-based THz SBDs were irradiated by 450 MeV ^84^Kr^18+^ beam with irradiation fluences of 5 × 10^6^ ions/cm^2^, 10^7^ ions/cm^2^, and 2.5 × 10^7^ ions/cm^2^, respectively. The Hall samples of n-GaN were also irradiated simultaneously.

The current–voltage (*I-V*) and capacitance–voltage (*C-V*) characteristics of the GaN-based THz SBD were measured by an Agilent B1500A semiconductor parameter analyzer and Keysight E4980A LCR meter, respectively. PL spectra were detected by a microregion spectrometer with a 325 nm He-Cd laser. LFN spectroscopy was performed using a Primarius FS-Pro system.

## 3. Results and Discussion

### 3.1. Effects of Irradiation on Electrical Properties

The room temperature *I-V* and *C-V* characteristics of the GaN-based THz SBDs were measured as shown in Figure 2a,b. GaN-based THz SBDs with and without SHI irradiation exhibit excellent Schottky diode properties, and their *C-V* characteristics are consistent with traditional Schottky junction depletion. Compared to pristine GaN-based THz SBDs, the SBDs with SHI irradiation exhibit similar forward *I-V* and *C-V* characteristics, but a slight increase in *I_R_*. The extracted *I_R_* values under varying irradiation fluences at −20 V are presented in Figure 2c. In SHI irradiation experiments, the standard irradiation fluence is typically 10^7^ ions/cm^2^. By linearly fitting the *I_R_* data of GaN-based THz SBDs at −20 V, the *I_R_* increases by 1.48 × 10^−7^ A for every 10^7^ ions/cm^2^ increase in Kr SHI irradiation fluence.

The *f_c_* at zero bias determines the operating frequency of GaN-based THz SBDs, which can be calculated based on the series resistance (*R_s_*) and total capacitance (*C*_0_), as follows [31,32]:(1)$$f_{c}=\frac{1}{2\pi R_{s}C_{0}}$$

The *R_s_* and *C*_0_ were extracted from forward *I-V* and *C-V* data, respectively, and the *f_c_* of GaN-based THz SBDs for varying irradiation fluences were calculated in Figure 2d. As the irradiation fluence increases, the *f_c_* of GaN-based THz SBDs decreases, with a 4% reduction in *f_c_* observed at Kr SHI irradiation fluence of 10^7^ ions/cm^2^. The continuous degradation of device performance was accompanied by a sustained increase in the irradiation fluence.

The primary transmission mechanism of forward current in Schottky diodes is the thermionic emission (TE) theory, and the forward *I-V* relationship is given by Equation (2) [33].(2)$$I={AA}^{*}T^{2}exp(-\frac{q\varphi_{B}}{kT})[\exp\left(\frac{q(V-IR_{s})}{nkT}\right)-1]$$

Here, *A*, *A**, *T*, *φ_B_*, *k*, and *n* are the anode contact area, Richardson constant (26.5 A·cm^−2^K^−2^) [34], temperature, Schottky barrier height, Boltzmann constant, and ideality factor, respectively. We measured the *I-V* and *C-V* data of 50 devices both before and after Kr SHI irradiation, the *n*, and *φ_B_* were extracted based on Equation (2), and the average parameter values of GaN-based THz SBDs are displayed in Table 1. Based on the impact of Kr SHIs of 2.5 × 10^7^ ions/cm^2^ fluence, the *f_c_* deteriorated by 9%, the *I_R_* has increased more than twice, the *n* has slightly increased, and *φ_B_* has slightly decreased; this was determined by comparing the device performance before and after irradiation. At higher doses of Kr SHI irradiation, radiation damage accumulates further, leading to continued degradation of the performance of GaN-based THz SBDs.

To analyze the performance of GaN-based THz SBDs by Kr SHI irradiation at 2.5 × 10^7^ ions/cm^2^ fluence, the degradation distribution of key parameters for 50 GaN-based THz SBDs before and after irradiation is statistically summarized in Figure 3. Δ*f_c_*, Δ*I_R_*, Δ*n*, and Δ*φ_B_* follow Gaussian distributions centered at 27 GHz, 3.7 × 10^−7^ A@ −20 V, 0.022, and 0.016 eV, respectively. The *n* and *φ_B_* of GaN-based THz SBDs exhibit minor changes under Kr SHI irradiation, which do not significantly impact the Schottky junction properties. However, the *f_c_* and *I_R_* of GaN-based THz SBDs show significant degradation, with Δ*f_c_* deteriorating by up to 50 GHz and Δ*I_R_* increasing by up to 3.5 times at −20V, which highlight the sensitivity of high-frequency parameters to Kr SHI irradiation.

Defects induced by Kr SHI irradiation in GaN-based THz SBDs can be repaired through alloying effects. Consequently, the GaN-based THz SBDs subjected to Kr SHI irradiation were annealed at room temperature for 90 days and at 300 °C for 5 h. Both annealing methods can restore the performance degradation of GaN-based THz SBDs caused by Kr SHI irradiation, including reduced *I_R_*, improved *φ_B_*, and reduced *n*.

The Kr SHI irradiation damage primarily degrades the performance of GaN-based THz SBDs by inducing carrier removal. To accurately obtain carrier information in GaN-based THz SBDs and mitigate the influence of edge effects associated with the small anode size on carrier concentration distribution, *C-V* measurements were conducted on GaN SBDs with a 200 μm anode diameter before and after Kr ion irradiation, as shown in Figure 4a. A reduction in capacitance is observed after exposure to 2.5 × 10^7^ ions/cm^2^ of Kr SHI irradiation. The net doping concentration (*N_D_*-*N_A_*) in the n^−^-GaN layer was calculated from the *C-V* curves using Equation (3); here, *N_D_* and *N_A_* are donor and acceptor concentrations in the n^−^-GaN layer, respectively; *ε_s_* is the permittivity of GaN.(3)$$N_{D}-N_{A}=\frac{2}{q\varepsilon_{s}A^{2}}\frac{1}{d\left(1/C^{2}\right)/dV}$$

Figure 4b illustrates the carrier concentration profile in the n^−^-GaN layer. The carrier concentration stabilizes at approximately 5 × 10^17^ cm^−3^, which is consistent with the doping concentration during the epitaxial growth of the n^−^-GaN layer. The carrier concentration at the bottom of the n^−^-GaN layer exhibits a rapid increase attributed to the diffusion from the n^+^-GaN layer during the high-temperature epitaxial growth process. By comparing the carrier concentration in the n^−^-GaN layer before and after irradiation, we observe a slight reduction in the net carrier concentration, decreasing from approximately 4.8 × 10^17^ cm^−3^ before irradiation to approximately 4.6 × 10^17^ cm^−3^ following irradiation. The observed decline in carrier concentration can be attributed to the formation of acceptor-type deep-level trap states induced by Kr SHI irradiation. These defects exhibit charge-compensating properties by trapping free carriers, ultimately leading to a reduction in the net carrier density within the n^−^-GaN layer.

### 3.2. Kr SHI Irradiation Simulation

The mechanisms of energy loss, 450 MeV Kr ion-irradiation distribution, stopping range, ionization, and displacement in GaN-based THz SBDs caused by Kr SHI impacts are analyzed using the SRIM software (SRIM-2013). SRIM is based on the Monte Carlo algorithm and employs a detailed full-damage cascades model to simulate and track the trajectories of a large number of incident particles in solid materials. The trajectories of 450 MeV Kr ions into the GaN-based THz SBDs were shown in Figure 5a. The Kr ions penetrate the anode metal and GaN epitaxial layer and stop in the SiC substrate. The penetration depth of Kr ions is approximately 28 µm. As illustrated in Figure 5b, ionization energy deposition accounts for over 99.9% of the total energy loss of 450 MeV Kr ions. The energy deposition of Kr ions varies through the layers of Au-Ni/GaN/SiC structures, with ionization energy losses ranging from 32 to 19 keV/nm at the Au-Ni/GaN interface and from 19 to 14 keV/nm at the GaN/SiC interface.

The energy transferred to the target atoms via elastic collisions between Kr ions and target atomic nuclei significantly exceeds the displacement damage threshold of target atoms, leading to the formation of vacancies due to atomic displacement. According to SRIM calculations, the collision of a Kr ion in GaN-based THz SBDs generates approximately 1534 vacancies. Figure 5c displays the vacancy densities of 0.2/nm in the GaN epitaxial layer and 0.1/nm in the SiC substrate by per Kr ion irradiation. The performance degradation of GaN-based THz SBDs caused by Kr ion irradiation is attributed to displacement damage within the GaN epitaxial layer. Consequently, Kr SHI irradiation introduces Ga vacancies (*V_Ga_*) at a concentration of 0.13/nm and N vacancies (*V_N_*) at 0.065/nm in the GaN epitaxial layers, as shown in Figure 5d. The high *V_Ga_* concentration is primarily attributed to the low threshold energy required for Ga atom departure in GaN materials.

### 3.3. The Analysis of Low-Frequency Noise

To investigate the influence of Kr SHI irradiation on the performance of GaN-based THz SBDs, the LFN spectra of the devices under a forward bias of 0.8 V were measured and are shown in Figure 6a. The observed reduction in LFN following Kr SHI irradiation is attributed to the decreased current. The LFN data at a forward bias of 0.8 V reveal a total noise level comprising superimposed Lorentzian humps and 1/*f* noise, as described by Equation (4) [35,36].(4)$$S_{I}=\frac{A\tau }{1+{\left(2\pi f\tau \right)}^{2}}+\frac{B}{f}$$
where *S_I_* is the LFN power spectral density. *A*, *B* are the amplitudes of Lorentzian hump and 1/*f* noise, respectively. τ is the time constant associated with the trap states.

The LFN spectra under different irradiation fluences were fitted using Equation (4), and the extracted values of *A*, *B*, and τ are summarized in Table 2. The trap energy level (*E_T_*) below the conduction band was calculated from the time constant τ using Shockley–Read–Hall statistics [37](5)$$\tau =\frac{1}{v_{th}\sigma_{T}N_{C}}exp(\frac{E_{T}}{kT})$$
where *σ_T_* = 1 × 10^−14^ cm^2^ is the capture cross-section of the trap states, *N_C_* = 4.3 × 10^14^ T^3/2^ cm^−3^ is the effective density of states in the conduction band, and *ν_th_* = 2.5 × 10^7^ cm/s is the average thermal velocity of the carriers [38]. The values of *E_T_* with and without Kr SHI irradiation are presented in Table 2, indicating that the trap states in GaN-based THz SBDs are located approximately 0.62 eV below the conduction band. Furthermore, as the fluence of Kr SHI irradiation increases, the *E_T_* becomes shallower.

Figure 6b displays the contrast of the LFN with current for GaN-based THz SBDs with and without Kr SHI irradiation. Compared to pristine GaN-based THz SBDs, the irradiated devices exhibited relatively lower LFN power. The LFN at low current follows a relationship of *S_I_*∝*I*^0.82^, whereas it changes to *S_I_*∝*I*^2^ as the current increases. The LFN of GaN-based THz SBDs under forward bias exhibits a strong dependence on the *R_s_*. At low current and high *R_s_* conditions, LFN is dominated by Lorentzian noise, which is primarily attributed to interface defects at the Schottky metal–semiconductor junction. At high current and low *R_s_*, changes in barrier height caused by the capture and release of electrons by defect states result in low-frequency 1/f noise [39,40].

### 3.4. The Analysis of Photoluminescence

It is well known that the *E_T_* of GaN material is verified by the yellow luminescence (YL) in PL spectra [41,42,43]. To further investigate the generation of traps in GaN-based THz SBDs induced by Kr SHI irradiation, PL spectra were measured, as shown in Figure 7. The PL spectra of GaN epitaxial wafers were normalized to the peak intensity of the GaN bandgap (3.4 eV), and the wafers exhibited the same PL spectra both before and after Kr SHI irradiation. However, the PL intensity of YL decreases with increasing Kr SHI irradiation fluence, indicating that defects are introduced by the Kr SHI irradiation.

The PL spectrum of GaN material includes multiple emission peaks, such as the red luminescence (RL) peak at 1.93 eV and the YL peak at 2.2 eV. To better investigate the effect of Kr SHI irradiation fluence on the PL characteristics of GaN material, two Gaussian functions were employed to fit the PL spectra of the GaN epitaxial layers both before and after Kr SHI irradiation, as presented in Figure 8. The RL and YL emissions involve a possible transition mechanism associated with the recombination of donor–acceptor pairs during trap energy-level transitions [44,45]. As shown in Figure 8a, unirradiated GaN epitaxial wafer exhibits RL and YL spectra, which are attributable to inherent defects within the GaN material. As illustrated in Figure 8b–d, the intensity of RL and YL increases with the rising Kr SHI irradiation fluence, which is attributed to the defect formation caused by irradiation damage.

The peak parameters of the RL and YL spectra are summarized in Table 3. The area ratio is defined as the ratio of the integrated intensity of defect band emission to that of bandgap emission. With the increasing Kr SHI irradiation fluence, the area ratio of the RL and PL emission bands increases, correlating with the observed rise in trap states in the LFN spectra. Combined with the studies about the impact of SHI irradiation on GaN under different irradiation fluence [25,26,27,30,33,36], these results further demonstrate the significant influence of SHI irradiation on the surface properties of GaN epitaxial layers, which can ultimately impact the performance of a device.

## 4. Conclusions

In summary, this work systematically investigated the effects of 450 MeV Kr SHI irradiation on the electrical characteristics and induced defects in GaN-based THz SBDs. It was found that the Kr SHI irradiation degraded the high-frequency performance of GaN-based THz SBDs due to irradiation-induced defects in the GaN epitaxial layer. After Kr SHI irradiation, decreases in *f_c_* and *φ_B_* were observed, accompanied by increases in *I_R_* and *n*. As confirmed by SRIM calculations, Kr SHI irradiation introduced displacements and vacancies, which significantly degraded the high-frequency performance of GaN-based THz SBDs. The trap states with an energy level approximately 0.62 eV below the conduction band were identified through LFN spectra. Furthermore, the analysis results based on RL and YL spectra confirm the presence of irradiation-induced defects caused by Kr SHI irradiation. These findings demonstrate that the high-frequency performance of GaN-based THz SBDs in the THz band is highly sensitive to heavy ion irradiation. Under prolonged irradiation conditions, irradiation-induced defects gradually accumulate, leading to continuous performance degradation and reduced durability of GaN-based THz SBDs.

## Figures and Tables

**Figure 1 micromachines-16-00288-f001:**
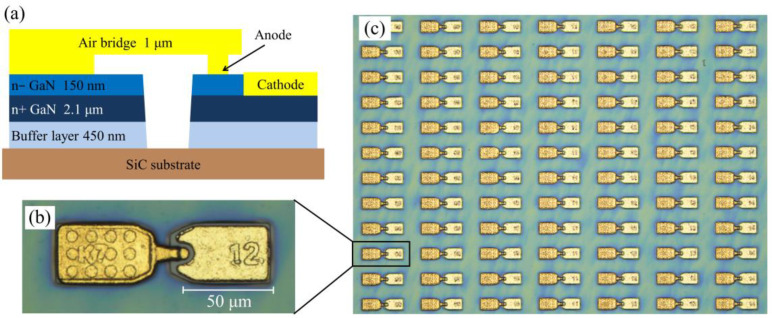
(**a**) Cross-sectional view, (**b**) image, and (**c**) distribution of GaN-based THz SBD.

**Figure 2 micromachines-16-00288-f002:**
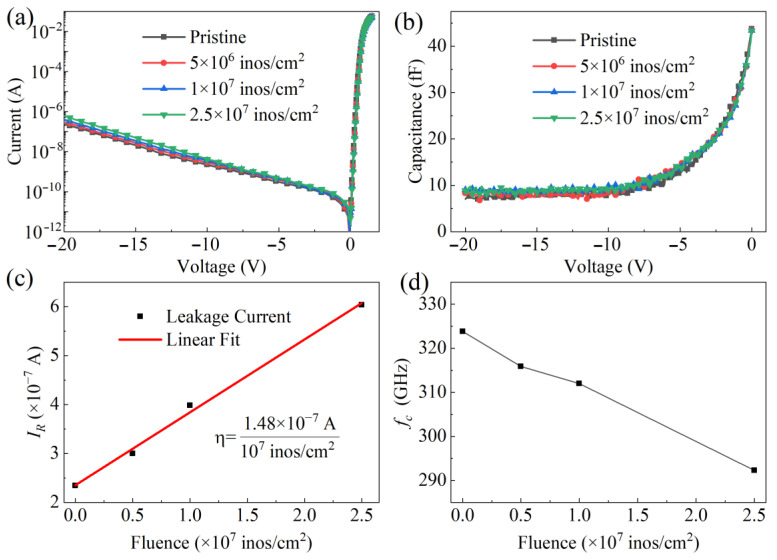
(**a**) *I-V* and (**b**) *C-V* characteristics of GaN-based THz SBDs at different fluences, (**c**) *I_R_* and (**d**) *f_c_* of GaN-based THz SBDs before and after Kr SHI irradiation.

**Figure 3 micromachines-16-00288-f003:**
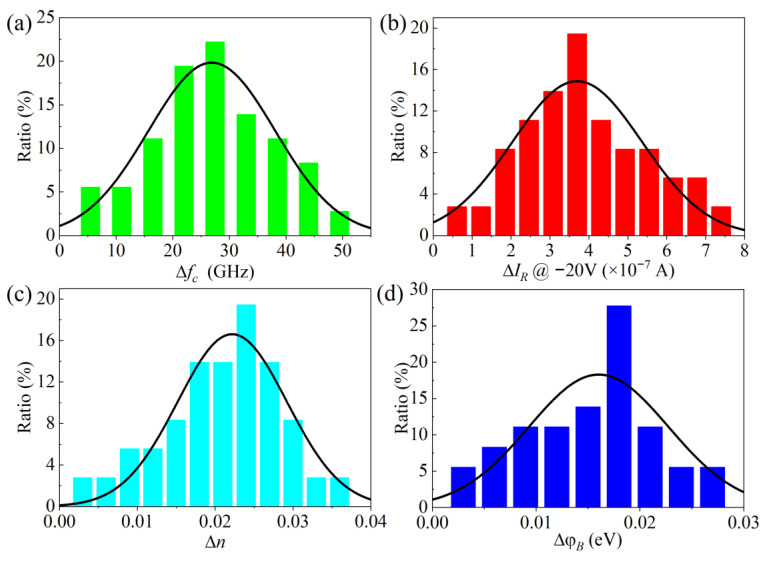
The degradation distribution of (**a**) Δ*f_c_*, (**b**) Δ*I_R_*, (**c**) Δ*n*, and (**d**) Δ*φ_B_* of 50 GaN-based THz SBDs before and after irradiation.

**Figure 4 micromachines-16-00288-f004:**
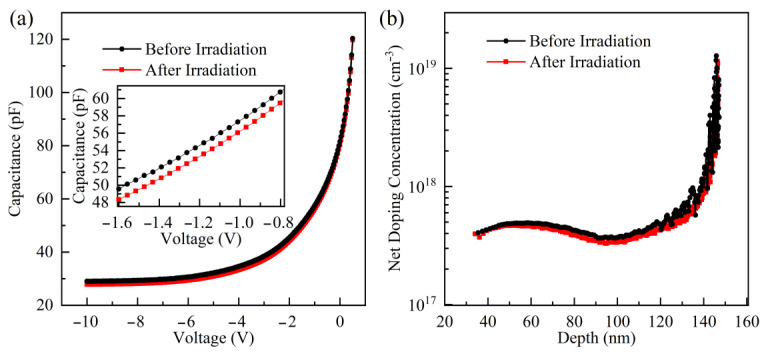
(**a**) *C-V* characteristics of GaN SBDs before and after irradiation, and (**b**) net doping concentration profile as a function of depth derived from these *C-V* measurements.

**Figure 5 micromachines-16-00288-f005:**
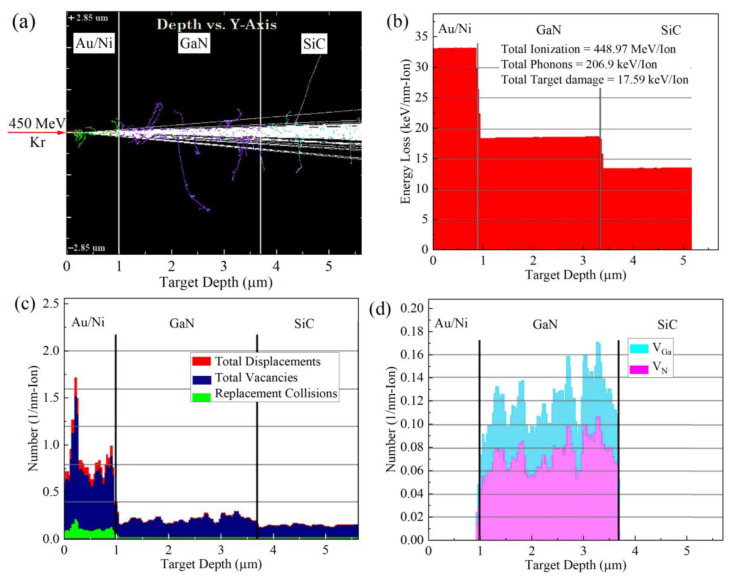
(**a**) Trajectories of 450 MeV Kr ions in GaN-based THz SBDs, (**b**) variation in electronic energy loss vs. ions’ path length in Au/Ni/GaN/SiC target, (**c**) distribution of displacement losses and (**d**) vacancy concentration (*V_Ga_* and *V_N_*) in the GaN epitaxial layer for 450 MeV Kr ions in GaN-based THz SBDs.

**Figure 6 micromachines-16-00288-f006:**
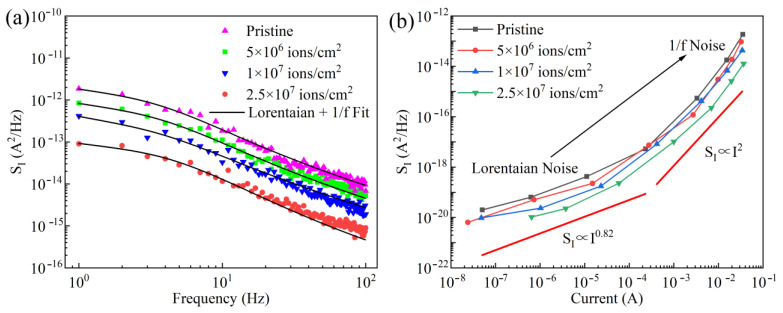
(**a**) The LFN spectra of GaN-based THz SBDs at 0.8 V. (**b**) Current dependences of LFN for GaN-based THz SBDs at 10 Hz.

**Figure 7 micromachines-16-00288-f007:**
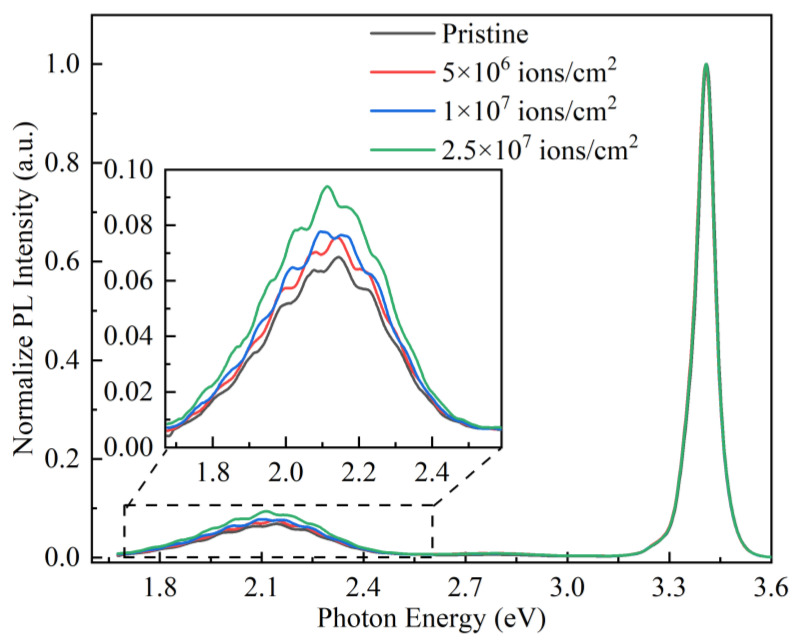
PL spectra of GaN-based THz SBDs with and without Kr SHI irradiation.

**Figure 8 micromachines-16-00288-f008:**
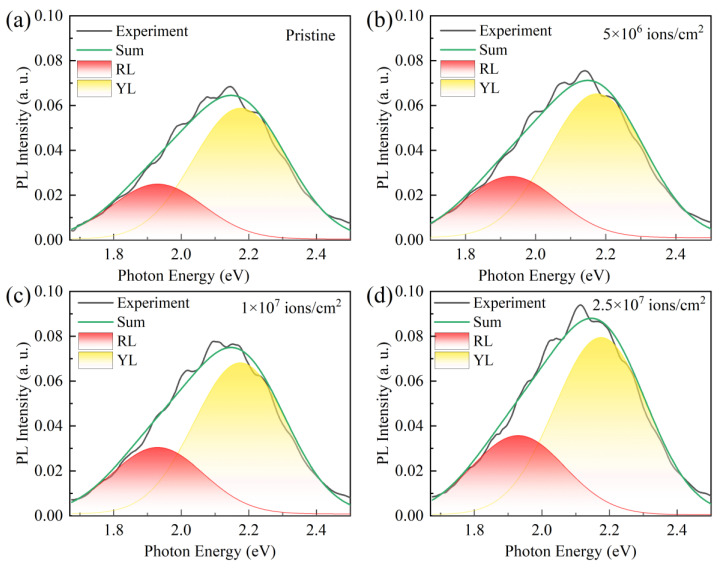
(**a**) The RL and YL spectra of (**a**) the pristine GaN epitaxial wafer and Kr SHI irradiated GaN epitaxial wafers with irradiation fluences of (**b**) 5 × 10^6^ ions/cm^2^, (**c**) 1 × 10^7^ ions/cm^2^ and (**d**) 2.5 × 10^7^ ions/cm^2^.

**Table 1 micromachines-16-00288-t001:** The key parameters of GaN-based THz SBDs with and without Kr SHI irradiation.

Fluences (Ions/cm^2^)	*f_c_* (GHz)	*I_R_* @ −20 V (A)	*n*	*φ_B_* (eV)
Pristine	320	2.3 × 10^−7^	1.20	0.69
2.5 × 10^7^	292	5.9 × 10^−7^	1.22	0.67

**Table 2 micromachines-16-00288-t002:** Extracted parameters for GaN-based THz SBDs with and without Kr SHI irradiation using Equation (4).

Fluence (Ions/cm^2^)	Pristine	5 × 10^6^	1 × 10^7^	2.5 × 10^7^
A (×10^−13^)	250	113	42	17
τ (ms)	46.7	43.6	42.9	37.4
B (×10^−13^)	8.0	3.9	2.5	0.35
*E_T_* (eV)	0.627	0.625	0.624	0.620

**Table 3 micromachines-16-00288-t003:** The parameters of RL and PL emission bands for the PL spectra in Figure 8.

	Emission Band	Photon Energy (eV)	Area Ratio (%)	Intensity (a. u.)
Pristine	RL	1.90	10.8	0.024
	YL	2.20	26.6	0.059
5 × 10^6^ ions/cm^2^	RL	1.93	11.9	0.028
	YL	2.21	28.8	0.065
1 × 10^7^ ions/cm^2^	RL	1.92	13.1	0.030
	YL	2.21	30.1	0.068
2.5 × 10^7^ ions/cm^2^	RL	1.93	15.8	0.036
	YL	2.21	35.3	0.080

## Data Availability

The data that support the finding of this study are available from the corresponding author upon reasonable request.

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
