# Peer review of "Effects of 450 MeV Kr Swift Heavy Ion Irradiation on GaN-Based Terahertz Schottky Barrier Diodes"

_micromachines, 2025, doi:10.3390/mi16030288_

Round 1

Reviewer 1 Report

Comments and Suggestions for Authors

The manuscript reports on fabricating and testing GaN-based terahertz (THz) Schottky barrier diodes under high-power 450 MeV Kr swift heavy ion (SHI) irradiation. SBD frequency performance was investigated in response to the irradiation damage induced in the GaN. The paper fills in the experimental gap by reporting on the high-energy SHI effect, however, its impact, in our opinion, will be limited because of the limited in-depth discussion. The paper repeats well-known and understood SHI effects of a wide range of irradiation energies induced defects in III-Ns. The presented data does not offer any new insight into the created defect type, defects capture-cross sections, defects accumulation/aggregation, defects migration, compensation etc. Because of that, we consider the paper to be a report on incremental rather groundbreaking results.

It is known that the response of GaN to radiation damage is a function of radiation type, dose, and energy, as well as the carrier density, impurity content, and dislocation density in the GaN. The latter can act as sinks for created defects and parameters such as the carrier removal rate due to trapping of carriers into radiation-induced defects depend on the crystal growth method used to grow the GaN layers. The growth method has a clear effect on radiation response beyond the carrier type and radiation source (ECS Journal of Solid State Science and Technology, 5 (2) Q35-Q60 (2016)). It is recommended that the authors comment on why the MOCVD-grown GaN was selected for the study.     

The authors shall comment on measured dependencies of noise on forward current and reverse breakdown voltage and estimate the contribution from the noise from the Schottky barrier and from the series resistance of the contacts etc.

Was the annealing effect of the swift ion beam considered in the result analysis?

Did/how the authors use the SRIM simulation to predict damage evolution? Please explain.

Show how the carrier concentration in the irradiated SBD correlates with the capacitance curves measured.  

Please comment on the defect built-up damage due to irradiation damage accumulation beyond the tested irradiation dose range. The device's durability resulting from prolonged high/low-dose irradiation should be commented on in the conclusion.

The manuscript requires English proofreading to eliminate typos.     

Author Response

Dear reviewer,

Thank you for the comments concerning our manuscript entitled “Effects of 450-MeV Kr swift heavy ion irradiation on GaN-based terahertz Schottky Barrier Diodes” (Manuscript ID: micromachines-3470114). These comments are all valuable and very helpful for revising and improving our paper. Please see the attachment.

Sincerely

Honghui Liu

Reviewer 2 Report

Comments and Suggestions for Authors

The authors submitted a research paper dealing with a GaN-based diode exposed to the ion irradiation. The research idea is not very original; however, the precise experiment and evaluation are good enough. Hence, the manuscript deserves to be published after slight revision that will follow the comments stated below:

(1) Defect identification:
Evaluated defect activation energy should be compared with reported defect energies (see GaN defect library https://doi.org/10.3390/inorganics12100257) to identify the origin and verify the simulation results (Fig.4(d)).

(2) Typos in order of magnitude:
The fluence is sometimes in the range of 107 ion cm-2 and sometimes 10-7 cm-2. The authors should avoid these typos! The plus/minus sign really matters.

(3) English:
The authors should proofread manuscript once again to avoid typos such as "inos" insterad of "ions" (Figure 2) or "Intersity" instead of "Intensity" (Table 3)

Comments on the Quality of English Language

The comments on English level were explained in the response to authors.

Author Response

Dear Reviewer,

Thank you for the comments concerning our manuscript entitled “Effects of 450-MeV Kr swift heavy ion irradiation on GaN-based terahertz Schottky Barrier Diodes” (Manuscript ID: micromachines-3470114). These comments are all valuable and very helpful for revising and improving our paper. Please see the attachment.

Sincerely

Honghui Liu

Round 2

Reviewer 1 Report

Comments and Suggestions for Authors

This is the revised manuscript. The authors addressed the reviewer’s comments, thank you, however not all responses are satisfactory. We recommend minor revisions before accepting for publication

The reply to Reviewer 1 Comment 1 is unsatisfactory. It is generic and does not offer any insight into the matter addressed in the original question/issue copied here: “It is known that the response of GaN to radiation damage is a function of radiation type, dose, and energy, as well as the carrier density, impurity content, and dislocation density in the GaN. The latter can act as sinks for created defects and parameters such as the carrier removal rate due to trapping of carriers into radiation-induced defects depending on the crystal growth method used to grow the GaN layers.“  We recommend, that the authors add relevant discussion.

The reply to Reviewer 1 Comment 4 is unsatisfactory. The author’s reply does not address the damage evolution issues at all. What can the reader learn about the damage evolution from the SRIM simulation for Kr swift heavy ion irradiation on GaN?    

Author Response

Dear reviewer,

Thank you for the comments concerning our manuscript entitled “Effects of 450-MeV Kr swift heavy ion irradiation on GaN-based terahertz Schottky Barrier Diodes” (Manuscript ID: micromachines-3470114). We thank you for allowing us to resubmit a revised copy of the manuscript and we highly appreciate your time and consideration.

Sincerely

Honghui Liu

Reviewer 1 Comment 1:

The reply to Reviewer 1 Comment 1 is unsatisfactory. It is generic and does not offer any insight into the matter addressed in the original question/issue copied here: “It is known that the response of GaN to radiation damage is a function of radiation type, dose, and energy, as well as the carrier density, impurity content, and dislocation density in the GaN. The latter can act as sinks for created defects and parameters such as the carrier removal rate due to trapping of carriers into radiation-induced defects depending on the crystal growth method used to grow the GaN layers.” We recommend, that the authors add relevant discussion.

Response:

Thank the reviewers for the comments.

The irradiation damage in GaN is determined by a complex interplay between irradiation parameters (type, energy, and dose) and material properties (carrier density, impurity content, and dislocation density), such as the carrier removal effect due to trapping of carriers into radiation-induced defects varies depending on the growth method of GaN layers. There are three common growth method used to grow GaN layers, namely molecular beam epitaxy (MBE), metal organic chemical vapor deposition (MOCVD), and, hydride vapor phase epitaxy (HVPE), each with its own characteristics in terms of defect density and distribution, as shown in Table 1. MOCVD technology is a widely used technique for epitaxial GaN growth, which are attributable to low inherent defect density, appropriate epitaxial growth rate, and cost-effectiveness. The low intrinsic defect density of GaN materials grown using this method can effectively mitigate the carrier removal effect caused by irradiation.

Table 1. The comparison of three GaN growth methods.

Growth method

Growth rate

Defect density

Radiation-induced damage

Cost

Carrier removal rate

MBE

low

ultralow

low

ultrahigh

low

MOCVD

middle

low

low

low

low

HVPE

high

high

high

low

high

The changes in manuscript:

Page 2, Line 28-41

“The irradiation damage in GaN is determined by a complex interplay between irradiation parameters (type, energy, and dose) and material properties (carrier density, impurity content, and dislocation density), such as the carrier removal effect due to trapping of carriers into radiation-induced defects varies depending on the growth method of GaN layers [18]. Metal organic chemical vapor deposition (MOCVD) technology is a widely used technique for epitaxial GaN growth, which are attributable to low inherent defect density, appropriate epitaxial growth rate, and cost-effectiveness. The low intrinsic defect density of GaN materials grown using this method can effectively mitigate the carrier removal effect caused by irradiation.” are added.

“This study adopts metalorganic chemical vapor deposition (MOCVD) technology for the epitaxial growth of GaN materials, leveraging its excellent radiation hardness, such as a low carrier removal rate, effective defect management, and hydrogen passivation effect [18]” are deleted.

Reviewer 1 Comment 4

The reply to Reviewer 1 Comment 4 is unsatisfactory. The author’s reply does not address the damage evolution issues at all. What can the reader learn about the damage evolution from the SRIM simulation for Kr swift heavy ion irradiation on GaN?

Response:

The mechanisms of energy loss, 450-MeV Kr ion-irradiation distribution, stopping range, ionization, and displacement in GaN-based THz SBDs caused by Kr SHI impacts are analyzed using the SRIM software. When Kr SHI impacts GaN-based THz SBDs, both ionizing energy loss (IEL) and non-ionizing energy loss (NIEL) occur in the GaN material. IEL serves as the primary mode of energy loss, while NIEL is the predominant mechanism responsible for displacement damage. The energy transferred to the GaN material via elastic collisions between Kr ions and GaN material significantly exceeds the displacement damage threshold of target atoms, leading to the formation of vacancies due to atomic displacement. A Kr SHI irradiation introduces Ga vacancies (VGa) at a concentration of 0.13/nm and N vacancies (VN) at 0.065/nm in the GaN epitaxial layers. The high VGa concentration is primarily attributed to the low threshold energy required for Ga atom departure in GaN materials. Simultaneously, SRIM simulations can also provide the following internal information regarding Kr SHI impacts on GaN-based THz SBDs: 1) the penetration depth of Kr ions is approximately 28 µm, 2) the energy deposition of Kr ions through the Au-Ni/GaN/SiC layers is 32 keV/nm, 19 keV/nm, and 14 keV/nm, respectively, 3) the collision of a Kr ion in GaN-based THz SBDs generates vacancy densities of 0.2/nm in the GaN epitaxial layer and 0.1/nm in the SiC substrate.

The changes in manuscript:

Page 6, Line 26-30

“The mechanisms of energy loss, 450-MeV Kr ion-irradiation distribution, stopping range, ionization, and displacement in GaN-based THz SBDs caused by Kr SHI impacts are analyzed using the SRIM software.” are added.

“To analyze the mechanisms of Kr ion range, energy loss, and ionization displacement, the incident process of 450-MeV Kr ions was calculated using SRIM/TRIM software.” are deleted.

Page 7, Line 18-21

“The high VGa concentration is primarily attributed to the low threshold energy required for Ga atom departure in GaN materials.” are added.
